# Engineering of *Bacillus* Promoters Based on Interacting Motifs between UP Elements and RNA Polymerase (RNAP) α-Subunit

**DOI:** 10.3390/ijms232113480

**Published:** 2022-11-03

**Authors:** Youran Li, Xufan Ma, Liang Zhang, Zhongyang Ding, Sha Xu, Zhenghua Gu, Guiyang Shi

**Affiliations:** 1Key Laboratory of Industrial Biotechnology, Ministry of Education, School of Biotechnology, Jiangnan University, Wuxi 214122, China; 2National Engineering Research Center for Cereal Fermentation and Food Biomanufacturing, Jiangnan University, 1800 Lihu Avenue, Wuxi 214122, China; 3Jiangsu Provincial Engineering Research Center for Bioactive Product Processing, Jiangnan University, 1800 Lihu Avenue, Wuxi 214122, China

**Keywords:** promoter engineering, *Bacillus licheniformis*, UP element, α-CTD

## Abstract

*Bacillus* genetics need more versatile promoters for gene circuit engineering. UP elements are widely distributed in noncoding regions and interact with the α-subunit of RNA polymerase (RNAP). They can be applied as a standard element for synthetic biology. Characterization of the binding motif between UP elements and RNAP may assist with rational and effective engineering. In this study, 11 *Bacillus* constitutive promoters were screened for strength in *Bacillus licheniformis*. The motif in UP elements from a strong native promoter, P_Lan_, was characterized. The influence of specific sequences on RNAP binding and expression strength was investigated both in vitro and in vivo. It was found that sequences up to 50 base pairs upstream of the consensus motif significantly contributed to α-CTD (the alpha subunit carboxy-terminal domain) association. Meanwhile, two repeats of a proximal subsite were able to more strongly activate the expression (by 8.2-fold) through strengthening interactions between UP elements and RNAP. Based the above molecular basis, a synthetic UP element, UP5-2P, was constructed and applied to nine wild-type promoters. Fluorescence polarization results demonstrated that it had an apparent effect on promoter–α-CTD interactions, and elevated expression strength was observed for all the engineered promoters. The highest improved core promoter, P_acpp_, was more strongly activated by 7.4-fold. This work thus develops a novel strategy for *Bacillus* promoter engineering.

## 1. Introduction

Well-evaluated promoters of diversified strength are crucial elements for complex in vivo expression in synthetic biology. To this end, they are believed not only to be standardized, but also to exhibit predictable and reliable activity under a variety of different conditions [1,2]. Strong heterologous overexpression has been achieved in *Escherichia coli* based on the specific interaction between the bacteriophage-derived, T7-specific RNA polymerase (RNAP) and the strong T7 promoter. Recent research concerning this Gram-negative bacterium has focused more on the engineering of native promoters to create highly regulated gene expression circuits [3,4,5]. Although multiple native promoters have been identified in *Bacillus*, fewer methods for their rational engineering have been shown to be highly efficient, compared with that of *E. coli* ones.

Structurally, bacterial promoters contain two recognition elements for RNAP and the assembled sigma factor: the classical −35/−10 hexamer and the UP element. The −35 and −10 elements, or the promoter core sequences, recognized by the sigma subunit of RNAP, determine to a great extent the strength of a promoter [6]. A vast majority of promoter engineering approaches are based on evolutionary or semi-rational strategies within these regions [7,8,9]. Robust libraries can be obtained through saturation mutagenesis or error-prone PCR. Then, a series of engineered promoters with broad expression profiles can be screened with the assistance of fluorescent proteins. The most frequently encountered issue here concerns the inability to produce a synthetic promoter with higher expression levels than a natural alternative. Accumulating experimental evidence has offered a detailed picture of the molecular mechanism underlying UP element-dependent promoter stimulation in *E. coli* [10]. Recently, Alper et al. provided an example of successfully engineered *E. coli* promoters in a study that employed designing and optimizing UP elements of the promoter, with strengths up to 9.4-fold higher than their wild-type sequences [6]. This UP element was reported to contact the alpha subunit carboxy-terminal domain (α-CTD) of the RNAP, thus exerting significant control on promoter activity in a manner that is independent of the promoter core sequences.

*Bacillus* RNAP also contains α2 (dimer), β, β′, and σ subunits [11]. The holoenzyme assembled with a major sigma factor σ^A^ (analogous to σ^70^ in *E. coli*) binds to promoters to initiate transcription. Kang et al. designed synthetic *Bacillus* promoters by shuffling the promoter core sequences recognized by different sigma factors [12]. A set of strong promoters were identified, with a strength up to 18.6-fold higher with respect to that of the commonly used promoter P43. Although current genetic and biochemical information has proved that the UP elements were involved in α-CTD binding, which is required for high expression of multiple *Bacillus* promoters, modifying promoter activity through the engineering of UP elements has received limited attention in the field. Here, we seek to engineer a previously verified α-CTD-binding *Bacillus* redox-sensitive promoter TrxA [13], based on interacting motifs between UP elements and α-CTD of RNAP. To do so, we first prepared purified protein of α-CTD and located the UP sequence of TrxA via DNase I footprinting and electrophoretic mobility shift assay (EMSA). Then, the native one was replaced by predicted UP elements from other promoters. The synthetic promoters were characterized both in vitro for DNA–protein interaction and in vivo for the strength of expression of Enhanced Green Fluorescent Protein (EGFP). The results were analyzed for correlation, and finally, four UP elements were proved to more strongly activate expression of the TrxA core promoter by up to 7.3-fold. This work thus develops a novel strategy for *Bacillus* promoter engineering through simply altering the UP elements alone in a manner that is independent of the core promoter itself.

## 2. Results

### 2.1. The Selected Promoters Vary in Strength and Stage-Specific Expression

Eleven promoters, both wild-type and artificial, were selected to be incorporated into a library for structural and functional comparison, listed in Table 1. Then, a fluorescence method was adopted to evaluate the strength and the trends over cultivation time. As the wild-type cells also exhibit false-positive signals, and the fluorescence may decay in cell cultures, the two issues that influence the results the most are the cell loading volume and sampling time for fluorescence measurements. To this end, a method for EGFP-assisted screening of the constitutive promoters in *B. licheniformis* was developed by using BlR2G as an example. Fluorescent signals from cells loaded with different concentrations were investigated. Results showed that an OD600 range of 0.8–2.5 in a single well of the 96-well microtiter plate guaranteed a linear correlation between cell counts and the fluorescent signals (Appendix A).

Then, the cell cultures corresponding to different promoters were sampled at 12, 24, 36, and 48 h, respectively, and relative fluorescence intensity (fluorescence intensity (Au)/OD600) was used to demonstrate the promoter activities (Figure 1). The selected 11 constitutive promoters showed a wide range of estimated activities, with relative fluorescence intensity being the highest (64896) in P_Lan_ at 48 h and the lowest (860) in P43 at 24 h. More importantly, comparison between different sampling times revealed huge variations in transcription initiation. For example, P_shuttle09_ was the most active promoter at both 12 and 24 h, with a relative fluorescence intensity of more than 2.3-fold higher than that of the other promoters. From 24 h to 36 h, however, P_Lan_ experienced a great boost in relative fluorescence intensity (by more than 13-fold). Its activity continued to increase by 2-fold from 36 h to 48 h. At the same time, 9 of 11 promoters showed a significantly decreasing trend in activity. When relating the above results to cell growth of the strains, we found that 24 h was near the division point between exponential and stationary, so that these promoters exhibited transcription initiation patterns corresponding to different cell growth stages. Finally, the estimated promoter activity of P_Lan_ could be as high as 75-fold higher when the most frequently used promoter P43 was used as a reference. This promoter was then selected for further analysis and engineering.

### 2.2. P_Lan_ Was Predicted to Be Involved in Quorum Sensing (QS)

In *B. licheniformis,* P_Lan_ exhibited much higher activity than the other ones tested. This promoter was mined and predicted based on our previous transcriptome data. We wondered why it mediated hyper-expression and if it could be a source of UP elements for promoter engineering. Therefore, its genome loci were retrieved to find the possible answers. The highly expressed transcript RS00925 mediated by P_Lan_ is located within a lantibiotic biosynthesis gene cluster and codes for the putative propeptide. The amino acid sequence of this peptide is close to the one reported in *B. subtilis* and other bacteria [22,23]. However, one additional gene cluster with an unknown phenotype was observed just upstream of the P_Lan_ promoter. Inside this cluster, four genes have been annotated as being signal transduction histidine kinase (LanC), transcriptional regulatory protein (LanB), and two hypothetical proteins (LanD and LanA). Subsequent sequence alignment indicated that those sequences were consistent with the agr QS system containing AgrBDCA, which was well characterized in Gram-positive *Staphylococcus aureus*. This is related to cell density-dependent control of behaviors such as virulence and biofilm formation [24,25,26]. Specifically, LanC shares 23% identity with AgrC, LanB shares 21% identity with AgrB, LanD shares 30% identity with AgrD, and LanA shares 24% identity with AgrA. Notably, highly conserved or similar amino residues could be observed in specific domains between these four pairs of proteins. As a result, we can conclude that P_Lan_ is involved in an agr-like QS system. Most QS systems possess the characteristic of autoinducer-dependent transcriptional activation to ensure hyper expression of QS operons. It has also been reported that C-terminal domain of the alpha subunit of RNA Polymerase (α-CTD) plays an important role in this activation by making contact with the promoter regions [27,28]. We therefore examined whether this is also true for the QS system predicted in *B. licneniformis*.

### 2.3. Heterologously Expressed α-CTD Directly Interacts with Both P_trxA_ and P_Lan_ In Vitro

α-CTD from *B. licheniformis* was expressed by using pET-28(a)/DE3 system in *E. coli*. After the construct EcCTD was cultivated and induced, cell lysate was subjected to SDS-PAGE (sodium dodecyl sulfate-polyacrylamide gel electrophoresis). Compared with the control one, the sample corresponding to EcCTD apparently showed a band of 15 kD, which is consistent with the predicted molecular weight of α-CTD (Figure 2). As a His-tag was fused into the 3′end of the recombinant protein, purified α-CTD could be easily obtained by using a Mag-Beads His-Tag Kit. Subsequent SDS-PAGE confirmed that the purified sample exhibited a single band of ~15 kD, and this sample can be used for in vitro experiments on interaction between DNA and protein. P_trxA_ has been reported to specifically contact with α-CTD through UP elements in *B. subtilis*. Therefore, this sequence was chosen as a reference DNA sample to examine whether the recombinant α-CTD was structurally positive. Then, EMSA was used to characterize the DNA–protein interaction. As can be seen from the chromatography, the migration of the 343-bp DNA fragment was retarded when it was incubated with α-CTD. In addition, a clear protein concentration response relationship was observed in the EMSA gel, within a range of 0.1–0.3 g/L. This indicated a correct fold recognition of recombinant α-CTD which could bind to the reference probe.

We therefore performed additional EMSA to investigate whether P_Lan_ can interact with α-CTD. The probe corresponding to P_Lan_ was first amplified by PCR with biotin labeled at the 5′-end. When we incubated the probe with increasing amounts of α-CTD (from 0.1 to 0.3 g/L), different shifted bands were observed. It could be inferred that the complex α-CTD-P_Lan_ was formed in a protein concentration-dependent manner, confirming that α-CTD specifically bound to the promoter P_Lan_.

### 2.4. α-CTD Binds to an UP Element 50 bp Upstream of TSS

It is known that in *B. subtilis*, α-CTD can interact with specific UP elements located in a region upstream of the P_trxA_ core promoter. This UP element is believed to act as an activating sequence for certain promoters and has been consistently shown to increase promoter strength by facilitating RNA polymerase association with DNA. To determine the precise α-CTD binding site or the UP element, DNase I footprinting assays were performed with 5′-6-carboxyfluorescein (6-FAM)-labeled probes containing a sequence upstream of P_Lan_ core promoter region. As can be seen in Figure 3, the DNA region from −57 to −85 was protected from DNase I digestion. It was predicted to be an UP element (named UPL) able to interact with α-CTD. This predicted UP element was further verified by EMSA performed with α-CTD and truncated P_Lan_, in which UPL was removed through overlapping PCR. As expected from DNase I footprinting data, no shifted band was obtained with the truncated P_Lan_, from which a positive UP element could be inferred. Compared with previously reported UP elements in P_trxA_ (terms as UPA here), UPL contained a high-consensus sequence of “AAGAAATAAT” and sheared adenine (A) and thymine (T) dinucleotide repeat tracts. This has been proven to be a crucial element for RNAP recognition and binding in *B. subtilis* [29]. However, there were significant differences between the two UP elements in terms of genomic location. For example, UPA is located in the −40 to −31 region. UPL, in comparison, is located at a position further upstream (−60) of the TSS. Their flanking sequences also share little similarity. The influence of this newly found UP element on promoter activity remains elusive and needs further investigation.

### 2.5. UP Elements with More Flanking Base Pairs Ensured Higher Promoter Activity

UP elements as expression activators have been tested in different bacteria [30]. It was suggested that sequences flanking the UP element could influence promoter activity through certain DNA secondary structures. We have proven that the newly found UP element shares a consensus motif of 10 bp with that in P_trxA_. On this basis, we then selected the truncated UP element fragments with different lengths in flanking base pairs as DNA probes, and evaluated their interactions with α-CTD in vitro and the performance on expression activity. In this effort, we sought to identify a stronger UP element based on the ability to amplify expression of the P_trxA_ core promoter. For in vitro investigation, Fluorescence polarization (FP) provides a nonradioactive approach for measuring protein–DNA interactions. An increase in molecular volume due to binding of a small fluorescent molecule to a protein impedes the fluorescent molecule’s rotational motion in solution and results in an increase in polarization. According to this principle, the interaction between α-CTD and the engineered promoters with different flanking base pairs in UP elements was indicated by the degree of polarization change (DPC). It showed a clear correlation between the lengths of motif-flanking sequences and the polarization values (Figure 4). As shifted polarization values suggested stronger interactions, conformational changes of the DNA probes potentially induced by α-CTD interaction were revealed. It was found that sequences up to 50 base pairs upstream of the consensus motif significantly contribute to α-CTD association (*p* < 0.0001).

To demonstrate the P_trxA_ promoter activities mediated by the above different UP elements, cells harboring recombinant vectors were cultivated and collected for promoter strength detection, by using the same fluorescence method described in promoter selection. Wild-type P_trxA_ promoter was used as the control. As shown in Figure 5, all the synthetic promoters with UP elements embedded upstream of −35 sites displayed stronger activities than the wild-type one. The strongest promoter exhibited significantly higher strength (up to 5.6-fold) than the control. Consistent with the results in vitro, strong positive correlations were seen between the lengths in flanking base pairs of UP elements and the promoter activities, indicating that UP elements with more flanking base pairs ensured higher promoter activity. UP5 exhibited a potential stem-loop structure consisting of palindromic and inverted repeat sequences between the proximal subsite and the distal subsite. This is consistent with the best-characterized bacterial UP element, the one in the rrnB P1 promoter, in which the subsites are located between positions −40 and −60 with respect to the transcription start site [31]. Both subsites have been reported to stimulate transcription with varying degrees. We also found that there is no base preference in UP5. This is unexpected as previously characterized UP elements were reported to be rich in AT or CG tracts to facilitate interaction with α-CTD [32]. These findings prompted us to evaluate synthetic UP elements to improve promoter activities based on UP5.

### 2.6. Repeated Specific Subsite in UP Elements Improved Promoter Activity

According to the existing literature, it is possible that placing UP element sequences in tandem could further activate gene expression. However, the specific role of proximal subsites and distal subsites in activating expression remains unclear. Thus, we first attempted to improve promoter activity by inserting repeated UP5 sequences in the wild-type UP element (the semi-rational strategy). Several synthetic promoters harboring 1 to 5 tandem repeats of intact UP5 upstream of P_trxA_ were constructed (Figure 6). In preliminary screening, the promoter–α-CTD interactions were tested in vitro by FP. The results showed that although the polarization values seemed positively correlated with the number of repeats, the DPC caused by α-CTD interaction did not show significant distinctions. An EGFP-assisted secondary screening also showed that no activated expression was obtained compared with that of the initial synthetic promoter P_trxA_-UP5. These results indicate that simple repeats of UP5 could not improve promoter activity by providing an extended interaction region for α-CTD to bind.

As α-CTD binds to UP elements at both the proximal subsite and the distal subsite, we hypothesized that each subsite could influence the initiation of transcription individually. It is therefore worth trying to engineer promoters using the respective subsite as an activator. To do so, we designed two sets of synthetic promoters with tandem repeats of “CAAAATAGC” (proximal) and “GGCATTTTG” (distal), inserted upstream of UP5 sequences in P_trxA_-UP5, respectively. After those synthetic promoters were synthesized, they were labeled and incubated with α-CTD. FP results exhibited a wide distribution for DPC. The most repeats (five) of the proximal subsite had the highest DPC of 52%, suggesting a significantly changed interaction. The same repeat number of the distal subsite also had a DPC of 35%. Seven in ten synthetic promoters exhibited polarization values corresponding to stronger binding. In terms of fluorescence intensity, only four in ten synthetic promoters were higher than that of the parent promoter with only a single copy of UP5. Remarkably, the strongest performing sequence was not the most repeats of R5 but R2 (UP5-2P) for the proximal subsite, which more strongly activated P_trxA_ expression by 8.2-fold. Likewise, R2 for the distal subsite also exhibited the highest expression intensity. These results indicate that repeats of the specific subsite within an UP element could improve the downstream promoter performance by facilitating promoter–α-CTD interactions. Excessively strong interactions between UP elements and RNAP could decrease gene expression.

### 2.7. Synthetic UP Elements Improved Activity of Varied Core Promoters

The above results motivated us to investigate whether the synthetic UP element, UP5-2P, could function independently to improve the performance of promoters other than P_trxA_, especially those without a known α-CTD binding site. For this purpose, UP5-2P was inserted upstream of the other nine constitutive promoters screened previously. Just as predicted, as shown in Figure 7, FP results demonstrate that UP5-2P had an apparent effect on promoter–α-CTD interactions, with DPC values ranging from 33 to 85%. Fluorescence measurements of the EGFP reporting gene corroborated the above results in that elevated intensities were observed for all the engineered constructs. The most improved core promoter, P_acpp_, was able to be more strongly activated by 7.4-fold. Interestingly, all these promoters have not been reported to have an α-CTD binding site other than P_trxA_ and P_spovG_. These results suggest that the function of UP5-2P is transferable when it is embedded upstream of multiple core promoters.

## 3. Discussion

The genus *Bacillus* respresents a group of important bacterial hosts for synthetic biology, especially for its excellent performance at an industrial scale [33]. However, the development of the *Bacillus* expression system lags far behind *E. coli* or yeast. This can be largely attributed to the lack of strong promoters. The most currently used promoters are wild-type ones and originated in *B. subtilis*, such as P43 and P_HpaII_. The strongest constitutive promoter reported in *B. subtilis* is a hybrid one, P_Shuttle-09_, whose activity was 8 times that of P43. Interestingly, this hybrid promoter was assembled artificially from two *B. licheniformis* genes: *ylyB* and *luxS* [34]. In our current study, P_Shuttle-09_ was first evaluated in *B. licheniformis*, and the results also suggested that it was stronger than most reported *Bacillus* promoters within the first 24 h of cultivation. However, the activities of P_spovG_ and P_Lan_ were highly significant after 24 h, and P_Lan_ exceeded P_Shuttle-09_ by more than 28-fold in activity after 48 h. These results indicated that in addition to activities, theses promoters exhibited various patterns of transcription initiation corresponding to specific stages of cell growth. Based on the time point with the highest fluorescent intensity, P_trxA_ and P_Shuttle-09_ can be classified as “exponential active promoters”and P_spovG_ and P_Lan_ as “stationary active promoters” in *B. licheniformis*. It has also been reported that the expression of a series of genes was significantly upregulated during the transition from exponential growth to the stationary stage [35]. This type of cellular response may be controlled by two-component systems (quorum-sensing systems or other transcriptional regulators) to cope with the environmental changes. The promoter P_Lan_ in this study was supposed to be involved in an uncharacterized quorum-sensing system. In addition to the obvious time-course in the changes of fluorescent signals, sequence alignment also suggested that the gene cluster transcribed from P_Lan_ may serve the functions of recognition and secretion of cell–cell signaling molecules. This is the first time a *B. licheniformis* quorum-sensing-based promoter has been described; it may be used as a strong expression element for synthetic biology. Unlike the exponential active pattern of most currently used promoters [36], the cell concentration-dependent promoter may contribute to the rational design of a synthetic circuit which better balances cell growth and target compound biosynthesis.

RNAP has two points of physical contact with a promoter: the core promoter sequence (-10 and -35 element) and the UP element. This study confirms that the interaction between the specific DNA sequences in the UP element and α-CTD domain of RNAP can be utilized for rational engineering of promoters in *Bacillus*. As emphasized by Phillips et al., RNAP binding to the UP element occurs independently of the other promoter element [37]. We showed that manipulation of a well-characterized α-CTD binding motif could enhance the binding of RNAP to the promoter, which contributed to an improved expression intensity. This is similar to the case of the tandem promoter strategy to increase the strength. Researchers anticipated that tandem repetitive promoters might trap plentiful RNAPs, which in turn enhances gene expression [38]. Perhaps the most important finding in this study is that two repeats of the proximal/distal subsites strengthened the expression of the promoter the most, although five repeats of them were characterized as binding RNAP more tightly. Urtecho measured the expression strength of promoters in a library containing over 10,000 constructs and postulated that addition of an UP element would serve a compensatory role for promoters with weak −10 and −35 elements [39]. Based on their data, Phillips et al. further modeled the relationship between expression strength and RNAP–promoter free energy Δ*E*_RNAP_ [37]. They suggested that sufficiently strong RNAP–promoter binding energy could decrease gene expression because RNAP was glued to the promoter and unable to initiate transcription. Although these results were obtained in *E. coli*, the conclusion is consistent with our findings in *B. licheniformis*. A moderately repeated element, UP5-2P, more strongly activated P_trxA_ expression by 8.2-fold. Moreover, the wide range of activation or inhibition seen in the tandem proximal/distal subsite constructs suggests that the UP element has a significant effect on promoter function. A standardized UP element can be developed as a plug-and-play regulatory element for genetic circuit designing, in which the expression levels of the target gene could be tuned to an appropriate input level for a certain pathway. A clear advantage of this type of synthetic element is its portable and modular characteristics. We showed that when UP5-2P was placed upstream of nine constitutive promoters, all of them showed increased expression strength. This may be because these promoters were weaker compared with P_Lan_, and they received the greatest benefit upon addition of the UP element. We propose that the inhibition effect of a strong UP element, such as UP5-2P, would also be general and efficient for multiple wild-type promoters.

Finally, this study shows that UP element manipulation alone is competitive against other methods of promoter engineering and can be applied to modulate the activity of promoters with different core regions. This is especially beneficial for *Bacillus* hosts, as it is now inconvenient to construct complex genetic circuits due to a limited range of promoter activity. As UP elements are also likely to be important components of promoters recognized by secondary sigma factors, it is hypothesized that these UP elements could activate core promoters corresponding to various other sigma factors to a predictable extent.

## 4. Materials and Methods

### 4.1. Media and Strain Cultivation

The bacterial strains and plasmids used in this study are listed in Table 2. The reagents and medium for *Bacillus* transformation were prepared according to Li [33]. *E. coli* and *Bacillus* were grown in terrific broth (TB) based on Xiao’s methods [40]. Ampicillin (100 μg/mL) was added when necessary to maintain the plasmids in *E. coli*. *Bacillus* transformants were grown with 10 μg/mL erythromycin or 20 μg/mL tetracycline. Cultivation was performed at 37 °C in Luria–Bertani (LB) broth unless otherwise stated.

### 4.2. Plasmid Construction

For the purpose of strength evaluation, the selected promoters reported from other *Bacilli* strains and the predicted promoters in this study were cloned into pHY300-PLK, together with the *egfp* gene. All pHY plasmid constructs are listed in Table 1, and the primers used for amplification are listed in Appendix A. First, an *egfp* fragment was cloned and inserted between the H*ind*III and S*al*I sites of pHY300-PLK. Then, promoters from different origins were assembled upstream of egfp at the *Hind*III site by using primer pairs of HpaII-F/R ~ R2-F/R. Specifically, P_HpaII_ was cloned using the plasmid pMA5 as the template. P_spovG_, P_ydzA_, P43, and P_TrxA_ were amplified from genomic DNA of *Bacillus subtilis* 168. P_glvA_, P_Lan_, P_DS_, and P_shuttle-09_ were amplified from genomic DNA of *B. licheniformis* B1391. Finally, P_acpp_ and P_r2_ were synthesized according to the genomic sequence of *B. megaterium* DSM319 and *B. amyloliquefaciens* XH7, respectively.

For construction of synthetic promoters with different flanking sequence lengths, UP0~UP5 were introduced by primer pairs UP0-F~UP5-F and GFP-R. These PCR reactions were performed by using the plasmid pTrxAGFP as the template. The amplified fragments were also assembled in plasmid pHY300-PLK and then transformed into *E. coli* JM109.

For the construction of synthetic promoters with a repeated specific subsite in the UP element, UP5-1P~UP5-5P and UP5-1D~UP5-5D were introduced by primer pairs UP5-1P-F~UP5-5P-F, UP5-1D-F~UP5-5D-F, and GFP-R. These PCR reactions were performed by using the plasmid pTrxAGFPUP5 as the template. The amplified fragments were also assembled in plasmid pHY300-PLK and then transformed into *E. coli* JM109.

For the construction of synthetic promoters with the engineered UP element, UP5-2P, upstream of different parent promoters, primer pairs UPHpaII-F ~ UPR2-F and GFP-R were used to incorporate the engineered UP element, by using plasmids pHpaIIGFP~pR2GFP as the templates, respectively. The amplified fragments were also assembled in plasmid pHY300-PLK and then transformed into *E. coli* JM109.

The gene encoding α-CTD in the core enzyme of RNA polymerase was amplified using primer pairs CTD-F/R and gnomic DNA of B1391. This was then cloned between *Bam*HI and *Sal*I sites of pET-28(a) and transformed into competent *E. coli* DE3. Standard cloning and *E. coli* transformations were performed according to Sambrook and Russell [41]. To transform plasmids into *B. licheniformis*, a method of electrotransformation was used [42]. PCR reactions used PhantaTM Super-Fidelity DNA Polymerase from Vazyme biotech (Nanjing, China) and followed supplier instructions; primers were purchased from Sangon Biotech (Shanghai, China). All restriction enzymes were purchased from New England Biolabs (Ipswich, MA, USA). Fermentas T4 DNA ligase samples were purchased from ThermoFisher Scientific (Waltham, MA, USA). Vectors were isolated using an AxyPrep Plasmid Miniprep Kit from Axygen biosciences (Corning, CA, USA), and DNA purification was performed with an Axygen Magnetic Beads DNA purification Kit (Corning, CA, USA).

### 4.3. Fluorescence Measurements

Overnight culture of the recombinant *B. licheniformis* involved inoculation in 30 mL of TB medium supplemented with 20 g/L glucose at an inoculation volume of 3%. This was then kept at 37 °C with orbital shaking at 250 rpm. The culture (1 mL) was sampled at different time points. Cells were collected by centrifugation at 9000 *g*, washed with 0.9% saline, and diluted to OD_600_ = 0.5–1.0. GFP fluorescence was measured (SparK plate reader, Tecan, Männedorf, Switzerland) using a 96-well microtiter plate at an excitation wavelength of 485 nm, emission wavelength of 535 ± 15 nm, and a gain value of 100. Average fluorescence and standard deviation were calculated from the geometric mean fluorescence values of technical triplicates.

### 4.4. Purification of Recombinant α-CTD

The strain EcCTD harboring pCTD was inoculated in 30 mL TB media. When the OD_600_ of the cell culture reached 0.8, IPTG was added at a concentration 0.5 mM. After 12 h of incubation, cells were harvested by centrifugation and washed twice with a phosphate buffer (pH 7.4). They were lysed using a Sonics VCX 750 (Sonics and Materials, Inc., USA) ultrasonic processor and the crude enzyme was subjected to affinity purification using Mag-Beads His-Tag (Sangon biotech, Shanghai, China), following the manufacture’s instruction.

### 4.5. Electrophoretic Mobility Shift Assays (EMSA)

EMSA was carried out following Zhang’s report [43], with some modifications. Primers labeled with biotin were synthesized by Sangon biotech (Shanghai, China) and are listed in Appendix A. They were used for amplification of DNA probes. Different concentrations of purified α-CTD proteins were added to the reaction system containing 10 nM of the biotin-labeled fragment. After incubated at 25 °C for 20 min, the reaction samples were subjected to EMSA gel and transferred to the film, with an ice bath maintained throughout the process. The transferred nylon film was then irradiated with ultraviolet light to crosslink the sample. Finally, the image was obtained by ECL luminescence at different time intervals using a ChemiDoc XRS+ gel imager (Bio-Rad, Hercules, CA, USA).

### 4.6. Fluorescence Polarization (FP)

DNA probes were amplified by PCR using FAM labeled primers (synthesized by Sangon biotech) and the different promoters as templates. All the primers are shown in Appendix A. Labeled probe (100 nM) was incubated with 30 μg purified α-CTD protein in a binding buffer (25 mM Tris-HCl, 3 mM NaCl, 3 mM MgCl_2_, and 0.1 mM DTT) at room temperature for 20 min according to the method of Zhang [43]. Then, the total volume was added to 100 μL using the same binding buffer, and the sample was subjected to a multifunctional enzyme marker (BioTek Instruments, Winooski, VT, USA) for polarization value measurement. In this experiment, an excitation wavelength of 485 nm and an emission wavelength of 528 nm were used. The degree of polarization change (DPC) was defined as follows:$$Degree\;of\;polarization\;change\left(\%\right)=\frac{probe\;polarization\;value-protein\;incubated\;probe\;polarization\;value\;}{probe\;polarization\;value}\times 100\%$$

### 4.7. DNaseI Footprinting

DNase I footprinting experiments were performed as described in a previous study [44]. The DNA probes were prepared by PCR using the different promoters as templates with the FAM-labeled primers (synthesized by Sangon biotech) (Appendix A). The purified PCR product was incubated with the indicated amounts of purified α-CTD proteins at 37 °C for 20 min prior to digestion with 0.02 U DNase I (New England Biolabs, Beverley, MA, USA) for 30 s. The cleavage reaction was stopped by adding the same volume of stop solution (200 mM NaCl, 30 mM EDTA, 1% SDS) followed by phenol extraction and EtOH precipitation. DNase I digestion reactions were analyzed by capillary electrophoresis in an ABI 3730xl DNA Analyzer (Applied Biosystems, Foster City, CA, USA) with PeakScanner software v1.0 (Applied Biosystems, Foster City, CA, USA).

### 4.8. Statistical Analysis

All experiments were performed at least three times, and the results were expressed as the means ± standard deviations (SDs). Statistical analyses were performed using Student’s t test and ANOVA followed by a Tukey’s post hoc test. *p* values of less than 0.05 indicated significant results.

## Figures and Tables

**Figure 1 ijms-23-13480-f001:**
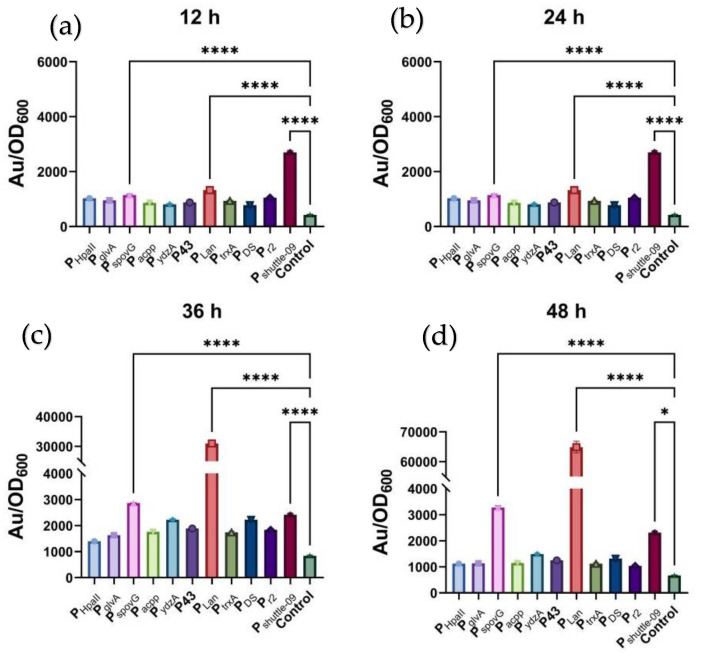
The strength and the trends over cultivation time of the selected promoters, characterized by the fluorescent method. (**a**) Relative fluorescence intensity at 12 h, (**b**) relative fluorescence intensity at 24 h, (**c**) relative fluorescence intensity at 36 h, (**d**) relative fluorescence intensity at 48 h. Au—Arbitrary unit. Results are expressed as mean of replicates. Columns with **** or * represent statistically significant differences between groups at *p* < 0.0001 or *p* < 0.05, respectively.

**Figure 2 ijms-23-13480-f002:**
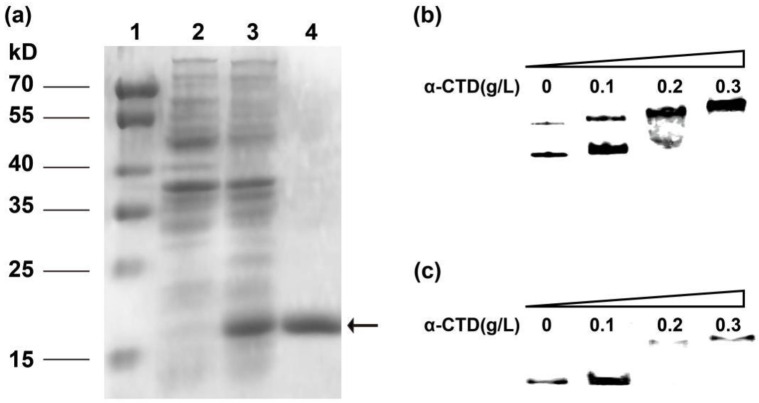
Purification of recombinant α-CTD and in vitro characterization of the interaction between α-CTD and P_trxA_/P_Lan_ by EMSA. (**a**) Recombinant α-CTD was purified by using affinity chromatography. Lane 1—standard molecular marker; lane 2—cell lysate of the control DE3 strain; lane 3—cell lysate of EcCTD; lane 4—purified α-CTD. The molecular weight of recombinant α-CTD is indicated by an arrow. (**b**) EMSA gels of α-CTD protein with increased loading of oligonucleotide probes corresponding to P_trxA_. (**c**) EMSA gels of α-CTD protein with increased loading of oligonucleotide probes corresponding to P_Lan_.

**Figure 3 ijms-23-13480-f003:**
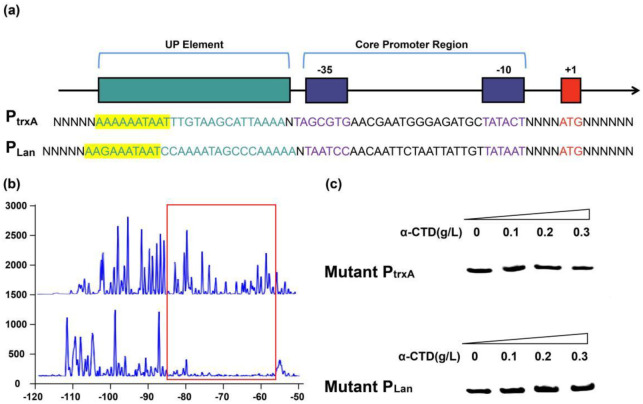
Characterization of α-CTD binding site of the promoters P_trxA_ and P_Lan_. (**a**) Illustration of the UP element and the core promoter region in the promoters P_trxA_ and P_Lan_. (**b**) α-CTD binding site was predicted by DNase footprinting in P_Lan_. (**c**) EMSA gels of α-CTD protein with oligonucleotide probes corresponding to mutant P_trxA_ and P_Lan_ with UPA or UPL truncated. The predicted binding sites of both promoters were thereby confirmed.

**Figure 4 ijms-23-13480-f004:**
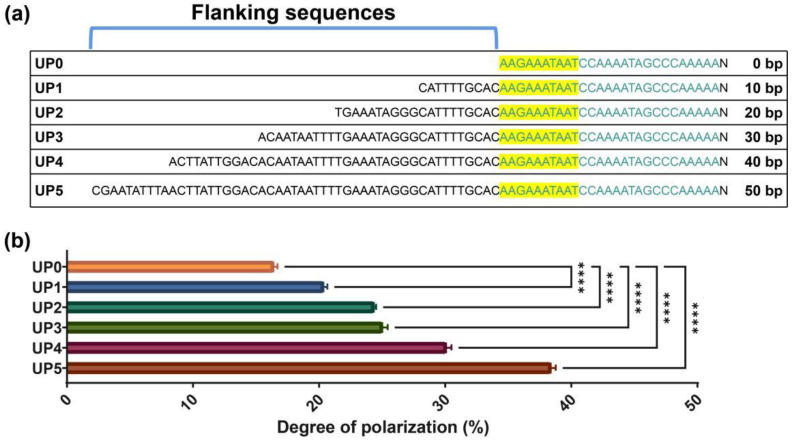
The influence of different flanking sequence lengths on the interaction between the promoters and α-CTD. (**a**) Illustration of the synthetic promoters with different flanking sequence lengths. (**b**) Interaction between the synthetic promoters and α-CTD were investigated by FP, and the degree of polarization was calculated. Results are expressed as mean of replicates; columns with **** represent statistically significant differences between groups at *p* < 0.0001.

**Figure 5 ijms-23-13480-f005:**
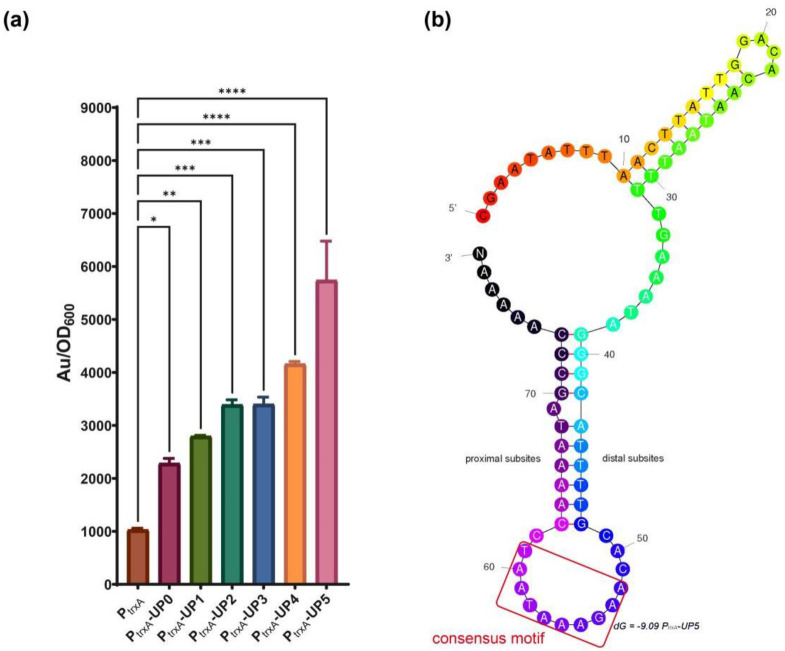
Expression strength of the synthetic promoters with different lengths in flanking base pairs. (**a**) Expression strength was compared by the florescent method, using wild-type P_trxA_ as the control. (**b**) The predicted structure of UPL. Au—Arbitrary unit. Results are expressed as mean of replicates; columns with ****, ***, ** or * represent statistically significant differences between groups at *p* < 0.0001, *p* < 0.001, *p* < 0.01 or *p* < 0.05, respectively.

**Figure 6 ijms-23-13480-f006:**
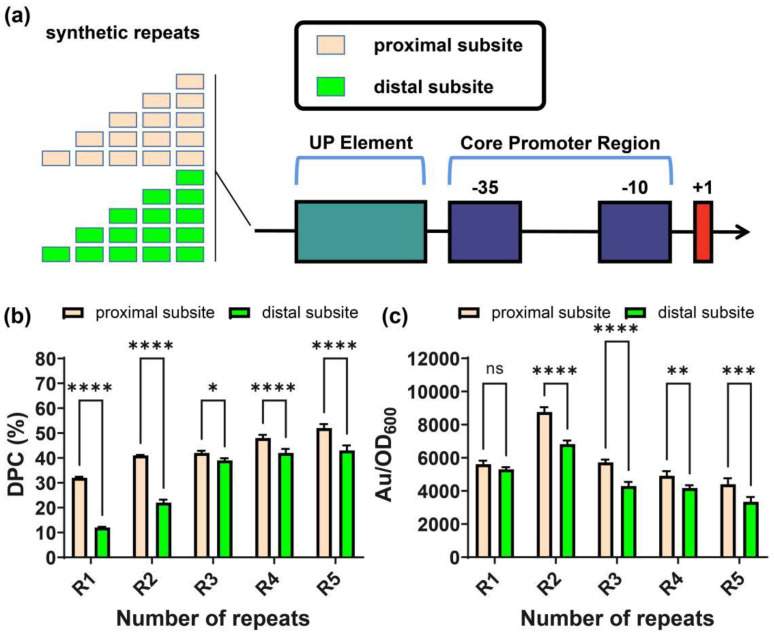
Illustration of the synthetic promoters and the characterization of their DPC and expression strength. (**a**) Different repeats of proximal/distal subsites were inserted upstream of the core promoter region of P_trxA_-UP5. (**b**) The DPC was investigated between synthetic promoters harboring different repeats of proximal/distal subsites. (**c**) The expression strength of the synthetic promoters was detected by the fluorescent method. Au—Arbitrary unit. Results are expressed as mean of replicates; columns with ****, ***, ** or * represent statistically significant differences between groups at *p* < 0.0001, *p* < 0.001, *p* < 0.01 or *p* < 0.05, respectively.

**Figure 7 ijms-23-13480-f007:**
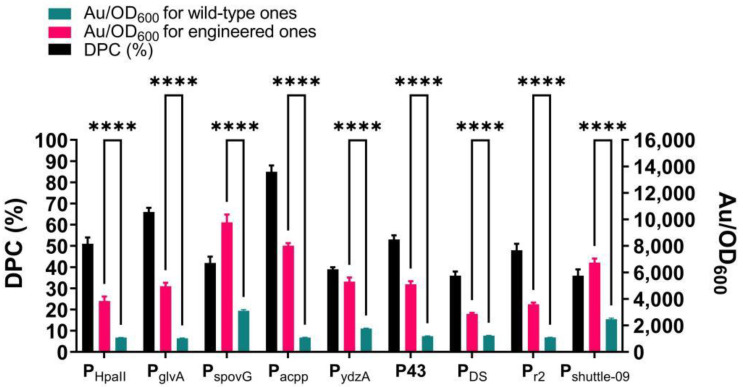
Characterization of DPC and expression strength of different promoters inserted with UP5-2P. Au—Arbitrary unit. Au is Arbitrary unit. Results are expressed as mean of replicates; columns with **** represent statistically significant differences between groups at *p* < 0.0001.

**Table 1 ijms-23-13480-t001:** The selected promoters of *Bacillus* origin.

Promoters	Description	Reference
P_HpaII_	Strong constitutive promoter in *B. subtilis*	[14]
P_glvA_	Upstream of *ynfA* in *B. licheniformis*	[15]
P_spovG_	Upstream of *spovG* in *B. subtilis*	[16]
P_acpp_	Upstream of *acpP* in *B. megaterium*	[17]
P_ydzA_	Upstream of *ydzA* in *B. subtilis*	[18]
P43	σ^55^ and σ^37^ promoter in *B. subtilis*	[19]
P_Lan_	Upstream of *lanC* in *B. licheniformis*	This study
P_TrxA_	Upstream of trxA in *B. subtilis*	[13]
P_DS_	Upstream of QGI43659.1 in *B. licheniformis*	This study
P_r2_	σ^w^ promoter in *B. amyloliquefaciens*	[20]
P_shuttle-09_	Artificial promoter in *B. subtilis*	[21]

**Table 2 ijms-23-13480-t002:** Bacterial strains and plasmids used in this study.

Strain or Plasmid	Description	Source
Strains		
*Escherichia coli* JM109	*F′, traD36, proAB +. lacIq,* *△* *(lacZ), M15/* *△* *(lac-proAB), glnV44, e14−, gyrA96, recA1, relA1, endA1, thi, hsdR17*	Our lab
*E. coli* BL21(DE3)	F-ompT gal dcm lon hsdSB (rB- mB) λ(DE3)	Our lab
*Bacillus licheniformis* CICIM B1391	Wild-type	Our lab
EcCTD	BL21, harboring pCTD	This work
BlHpaIIG	*B. licheniformis* CICIM B1391 harboring pHpaIIGFP	This work
BlSpovGG	*B. licheniformis* CICIM B1391 harboring pSpovGGFP	This work
BlYdzAG	*B. licheniformis* CICIM B1391 harboring pYdzAGFP	This work
Bl43G	*B. licheniformis* CICIM B1391 harboring p43GFP	This work
BlTrxAG	*B. licheniformis* CICIM B1391 harboring pTrxAGFP	This work
BlGlvAG	*B. licheniformis* CICIM B1391 harboring pGlvAGFP	This work
BlLanG	*B. licheniformis* CICIM B1391 harboring pLanGFP	This work
BlDSG	*B. licheniformis* CICIM B1391 harboring pDSGFP	This work
Bl09G	*B. licheniformis* CICIM B1391 harboring p09GFP	This work
BlAcppG	*B. licheniformis* CICIM B1391 harboring pAcppGFP	This work
BlR2G	*B. licheniformis* CICIM B1391 harboring pR2GFP	This work
BlTrxAGUP0	*B. licheniformis* CICIM B1391 harboring pTrxAGFPUP0	This work
BlTrxAGUP1	*B. licheniformis* CICIM B1391 harboring pTrxAGFPUP1	This work
BlTrxAGUP2	*B. licheniformis* CICIM B1391 harboring pTrxAGFPUP2	This work
BlTrxAGUP3	*B. licheniformis* CICIM B1391 harboring pTrxAGFPUP3	This work
BlTrxAGUP4	*B. licheniformis* CICIM B1391 harboring pTrxAGFPUP4	This work
BlTrxAGUP5	*B. licheniformis* CICIM B1391 harboring pTrxAGFPUP5	This work
BlTrxAGUP5P1	*B. licheniformis* CICIM B1391 harboring pTrxAGFPUP5P1	This work
BlTrxAGUP5P2	*B. licheniformis* CICIM B1391 harboring pTrxAGFPUP5P2	This work
BlTrxAGUP5P3	*B. licheniformis* CICIM B1391 harboring pTrxAGFPUP5P3	This work
BlTrxAGUP5P4	*B. licheniformis* CICIM B1391 harboring pTrxAGFPUP5P4	This work
BlTrxAGUP5P5	*B. licheniformis* CICIM B1391 harboring pTrxAGFPUP5P5	This work
BlTrxAGUP5D1	*B. licheniformis* CICIM B1391 harboring pTrxAGFPUP5D1	This work
BlTrxAGUP5D2	*B. licheniformis* CICIM B1391 harboring pTrxAGFPUP5D2	This work
BlTrxAGUP5D3	*B. licheniformis* CICIM B1391 harboring pTrxAGFPUP5D3	This work
BlTrxAGUP5D4	*B. licheniformis* CICIM B1391 harboring pTrxAGFPUP5D4	This work
BlTrxAGUP5D5	*B. licheniformis* CICIM B1391 harboring pTrxAGFPUP5D5	This work
BlHpaIIGUP5P2	*B. licheniformis* CICIM B1391 harboring pHpaIIUP5P2GFP	This work
BlSpovGGUP5P2	*B. licheniformis* CICIM B1391 harboring pSpovGUP5P2GFP	This work
BlYdzAGUP5P2	*B. licheniformis* CICIM B1391 harboring pYdzAUP5P2GFP	This work
Bl43GUP5P2	*B. licheniformis* CICIM B1391 harboring p43UP5P2GFP	This work
BlGlvAGUP5P2	*B. licheniformis* CICIM B1391 harboring pGlvAUP5P2GFP	This work
BlDSGUP5P2	*B. licheniformis* CICIM B1391 harboring pDSUP5P2GFP	This work
Bl09GUP5P2	*B. licheniformis* CICIM B1391 harboring p09UP5P2GFP	This work
BlAcppGUP5P2	*B. licheniformis* CICIM B1391 harboring pAcppUP5P2GFP	This work
BlR2GUP5P2	*B. licheniformis* CICIM B1391 harboring pR2UP5P2GFP	This work
Plasmids		
pMD19-T	*E. coli* cloning vector, Ap^R^	TaKaRa
pMA5	*E. coli/Bacillus* shuttle vector, Neo^R^/Ap^R^, *P_HpaII_*	Our lab
pHY300-PLK	*E. coli*/*Bacillus* shuttle vector, Ap^R^/Tet^R^	Our lab
pET-28(a)	*E. coli* expression vector, *Kan^R^*	TaKaRa
pCTD	pET-28a derivative with α-CTD encoding gene in *B. licheniformis*	This work
pHpaIIGFP	pGFP derivative with promoter P_HpaII_	This work
pSpovGGFP	pGFP derivative with promoter P_SpovG_	This work
pYdzAGFP	pGFP derivative with promoter P_YdzA_	This work
p43GFP	pGFP derivative with promoter P_43_	This work
pTrxAGFP	pGFP derivative with promoter P_TrxA_	This work
pGlvAGFP	pGFP derivative with promoter P_GlvA_	This work
pLanGFP	pGFP derivative with promoter P_Lan_	This work
pDSGFP	pGFP derivative with promoter P_DS_	This work
p09GFP	pGFP derivative with promoter P_shuttle-09_	This work
pAcppGFP	pGFP derivative with promoter P_acpp_	This work
pR2GFP	pGFP derivative with promoter P_r2_	This work
pTrxAGFPUP0	pTrxAGFP derivative with UP0 inserted	This work
pTrxAGFPUP1	pTrxAGFP derivative with UP1 inserted	This work
pTrxAGFPUP2	pTrxAGFP derivative with UP2 inserted	This work
pTrxAGFPUP3	pTrxAGFP derivative with UP3 inserted	This work
pTrxAGFPUP4	pTrxAGFP derivative with UP4 inserted	This work
pTrxAGFPUP5	pTrxAGFP derivative with UP5 inserted	This work
pTrxAGFPUP5P1	pTrxAGFPUP5 derivative with 1 proximal subsite inserted	This work
pTrxAGFPUP5P2	pTrxAGFPUP5 derivative with 2 proximal subsites inserted	This work
pTrxAGFPUP5P3	pTrxAGFPUP5 derivative with 3 proximal subsites inserted	This work
pTrxAGFPUP5P4	pTrxAGFPUP5 derivative with 4 proximal subsites inserted	This work
pTrxAGFPUP5P5	pTrxAGFPUP5 derivative with 5 proximal subsites inserted	This work
pTrxAGFPUP5D1	pTrxAGFPUP5 derivative with 1 distal subsite inserted	This work
pTrxAGFPUP5D2	pTrxAGFPUP5 derivative with 2 distal subsites inserted	This work
pTrxAGFPUP5D3	pTrxAGFPUP5 derivative with 3 distal subsites inserted	This work
pTrxAGFPUP5D4	pTrxAGFPUP5 derivative with 4 distal subsites inserted	This work
pTrxAGFPUP5D5	pTrxAGFPUP5 derivative with 5 distal subsites inserted	This work
pHpaIIUP5P2GFP	pGFP derivative with promoter P_HpaII-UP5-2P_	This work
pSpovGUP5P2GFP	pGFP derivative with promoter P_SpovG-UP5-2P_	This work
pYdzAUP5P2GFP	pGFP derivative with promoter P_YdzA-UP5-2P_	This work
p43UP5P2GFP	pGFP derivative with promoter P_43-UP5-2P_	This work
pGlvAUP5P2GFP	pGFP derivative with promoter P_GlvA-UP5-2P_	This work
pDSUP5P2GFP	pGFP derivative with promoter P_DS-UP5-2P_	This work
p09UP5P2GFP	pGFP derivative with promoter P_shuttle-09-UP5-2P_	This work
pAcppUP5P2GFP	pGFP derivative with promoter P_acpp-UP5-2P_	This work
pR2UP5P2GFP	pGFP derivative with promoter P_r2-UP5-2P_	This work

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
