# Peer review of "Engineering of Bacillus Promoters Based on Interacting Motifs between UP Elements and RNA Polymerase (RNAP) α-Subunit"

_ijms, 2022, doi:10.3390/ijms232113480_

Round 1

Reviewer 1 Report

This manuscript is of significance to a wide audience, in particular those working in synthetic biology.

I consider this manuscript suitable for publication, after the authors address the minor issues outlined below:

-          Line 20: write what α-CTD means.

-          Line 71: write in vitro and in vivo in italic.

-          Line 72: “The results…” instead of “Their results…”.

-          Line 82: “fluorescence” instead of “fluorescent”.

-    Line 92: “(Figure 1)” instead of “(figure 1)”, to be consistent with the remaining references to the figures.

-          Line 143: write in vitro in italic.

-   Line 159: despite here you show figure 2, this figure is not nowhere mentioned in the text (before or after the figure). When I analyze this image, I cannot find what you state above from lines 136 to 145. Where is the 15kDa band you refer in the text as the “the sample corresponding to PETD”? Please correlate the text and the figure.

-          Line 201: write what FP means.

-    Line 245: “(Figure 6)” instead of “(figure 6)”, to be consistent with the remaining references to the figures.

-          Line 291: with figure 7 the same happens as figure 2, there is no mention in the text.

Author Response

Thank you for your generous and positive comments.

Reviewer 2 Report

This work is generally well designed and can be considered to be published.

Author Response

(The authors gave the same response as above.)

Reviewer 3 Report

The reviewer has a few comments and suggestions which may help the authors to improve their report. These are listed in the attached MS Word file. 

Author Response

(The authors gave the same response as above.)
